# Refuge distributions and landscape connectivity affect host-parasitoid dynamics: Motivations for biological control in agroecosystems

Lucas D. Fernandes[1,2]*, Angelica S. Mata[3], Wesley A. C. Godoy[2], Carolina Reigada[4]

**1** Department of Life Sciences, Imperial College London, Silwood Park, Ascot, Berkshire, United Kingdom, **2** Departamento de Entomologia e Acarologia, Escola Superior de Agricultura Luiz de Queiroz—Universidade de São Paulo (USP), Piracicaba, SP, Brazil, **3** Departamento de Física, Universidade Federal de Lavras (UFLA), Lavras, MG, Brazil, **4** Departamento de Ecologia e Biologia Evolutiva, Universidade Federal de São Carlos (UFSCar), Rodovia Washington Luís, São Carlos, SP, Brazil

* ldiasfer@imperial.ac.uk

**Data Availability Statement:** All relevant data are within the paper and its Supporting information files.

## Abstract

Species distributions are affected by landscape structure at different spatial scales. Here we study how the interplay between dispersal at different spatial scales and landscape connectivity and composition affect local species dynamics. Using a host-parasitoid model, we assessed host density and host occupancy on the landscape, under different parasitoid dispersal ranges and three local distributions of non-crop habitats, areas where hosts are unable to grow but parasitoids are provided with alternative hosts and food resources. Our results show distinct responses of host density to increases in non-crop area, measured by differences in slopes for different distributions of non-crop habitats, and that the effect of local landscape composition on species dynamics depends on the landscape connectivity at the regional scale. Moreover, we show how host density and occupancy are affected by increasing parasitoid dispersal ranges depending on landscape structure. Our results demonstrate the role of local and regional scales on species distributions and the importance of the combined effects of species biological parameters and landscape structure on species dynamics. Finally, we highlight the relevance of these aspects for the development of better strategies of biological control.

## Introduction

The importance of landscape structure in influencing species distributions and metapopulation dynamics has been recognized by both theoretical and experimental studies [1–4]. Although it has been shown that heterogeneous landscape structure and specific dispersal patterns can alter population viability and species coexistence [5, 6], studies usually focus at a particular scale of spatial organization. Since there is now growing empirical evidence that species dynamics are in many cases subject to multi-scale processes [7–9], with important drivers

**Funding:** This work was partially supported by the Brazilian agencies FAPESP (https://fapesp.br/), CAPES (https://www.gov.br/capes/pt-br), FAPEMIG (http://www.fapemig.br/pt/), and CNPq (https://www.gov.br/cnpq/pt-br). L.D.F., W.A.C.G. and C.R. acknowledge funding provided by the São Paulo Research Foundation (FAPESP - grants \#2015/26989-4, \#2014/16609-7, and \#2019/11672-6). A.S.M. acknowledges computational time at DFI-UFLA and support by FAPEMIG (Grant No. APQ-02482-18) and CNPq (Grant No. 423185/2018-7). The funders had no role in study design, data collection and analysis, decision to publish, or preparation of the manuscript.

**Competing interests:** The authors have declared that no competing interests exist.

from local to regional scales, the question of how these different scales simultaneously affect species distributions still needs to be resolved.

This question is particularly relevant for agriculture, as landscape composition and structure, by conditioning movement patterns, directly influence density and occupation of insects and potential agricultural pests [10–13]. Furthermore, the definition of biological control strategies strongly depends on how landscape connectivity and local resource quality influence the dynamics of interaction between pests and control agents [14, 15].

Hymenopterous parasitoids are important control agents of economically relevant pests in both agricultural and natural environments. These insects have foraging behaviours that vary according to landscape composition, which influences their distribution and consequently the regulation of host populations [16, 17]. Generalist parasitoids can also play an important role on pest management, as their presence in landscapes combining crop and non-crop areas may help controlling pest populations [18]. When the insect pest is not present or present in low densities on the crop, the existence of alternative plants can promote refuge habitat for parasitoids, allowing their persistence in the area and collaborating to the prevention of future pest irruptions [19].

Many studies have shown the positive effects of habitat heterogeneity on the increase of abundance and diversity of natural enemies in agricultural ecosystems [12, 20, 21]. In this context, the management of crop areas to increase the heterogeneity of agricultural landscapes is a viable alternative in conservation biological control programs. Non-crop habitats support predators and parasitoids providing alternative prey/hosts and food resources, shelter, and overwintering areas, while also facilitating movement between different localities [22, 23]. It has been shown that increased percentages of non-crop areas may lead to increased rates of parasitism and reduced rates of herbivory [24]. However, the spatial distribution of this non-crop vegetation, not only its percentage, is a relevant aspect for biological control, since parasitism rates can also be affected by the distance between local host populations [25]. Thus, appropriate landscape compositions may promote the balance between effective pest control and natural enemies conservation [26, 27].

In the attempt to understand how different landscape features influence species interactions and distributions, mathematical and computational models have long been recognised as important tools [28]. Considering more specifically host-parasitoid interactions, many studies were conducted to analyze stability [29–31], persistence [31] and spatial dynamics in parasitoid communities [32]. Although these studies have significantly contributed to our understanding of spatial aspects of host-parasitoid models, none of them investigated the effects of simultaneously considering different spatial scales on insect movement, nor its implications for the effectiveness of biological control in agroecosystems.

Here we investigate how the interplay between dispersal at different spatial scales and landscape structural connectivity and composition (in other words, physical configurations and spatial features) affect species dynamics and distributions. We propose a mathematical model to investigate the dynamics of host-parasitoid populations in a space characterized by two distinct scales of species dispersal. On the local scale, species interact on a landscape with regular structure and short distances between patches. On the regional scale, different landscapes are interconnected by dispersal through comparatively larger distances. Each of these landscapes is characterized by different distributions of non-crop habitats, where hosts are unable to grow and parasitoids are provided with alternative hosts. We call these habitats parasitoid refuges, directly influencing parasitoids' spatial distributions and, consequently, their interaction dynamics with the primary hosts.

Specifically, we aim to assess (i) the direct impact of increasing proportions of parasitoid refuge areas on the host variables of density and occupancy, (ii) the differential effect of spatial

distributions of these refuge areas on the host variables, and (iii) how regional connectivity patterns affect species dynamics. Additionally, we also evaluate the effect of the parasitoid dispersal radius on the host variables at different spatial scenarios.

## Methods

### Spatial structure

In our model, space is represented by two layers of networks, corresponding to two distinct scales, which we name external network and square lattice (or internal network) as we show in Fig 1.

The external network represents the regional scale of an agricultural landscape, where each of the nodes corresponds to individual farms or production units, possibly linked. The links represent the possibility of dispersal of individuals between these units and, therefore, the network structure reflects the pattern of regional connectivity. We have used two different connectivity patterns for the external networks (Fig 1): (i) each node is linked only to its immediately adjacent neighbors (ring network), and (ii) several peripheral nodes are exclusively linked to one central node (star network). For the ring network, all the nodes are identical in relation to regional connectivity, whereas for the star network there is a clear separation in two groups (peripheral and central nodes).

The choice of these two arrangements highlights different spatial scenarios that might be imposed by regional topography: homogeneous connectivities, where all production units have the same number of neighbors, and heterogeneous connectivities, where the central unit behaves as a hub and the other units have only one link. The space between these units can represent urban areas, freshwater or any form of land use not suitable for the species we consider. While admittedly a simplification of reality, this representation enables the exploration of how differences in the structural connectivity of production units influence the dynamics of pests and natural enemies locally, emphasising the importance of integrated regional approaches to deal with common threats.

The internal networks describe the local structures of each of the production units and are presented as regular square lattices of size $L$ x $L$. Each of the nodes on the square lattice represents a patch, formed by a small group of individual plants. The dispersal of individuals between different patches depends on the dispersal scale of each species (see description below).

Patches on the lattice can be of two types: crop or non-crop areas. We assume hosts to be specialists on that specific crop and, therefore, unable to grow on non-crop areas. Additionally, non-crop areas are also assumed to provide alternative hosts and food supplies for parasitoids, allowing reproduction and maintenance of small populations, and we refer to them as refuges. Note that this is different from the more common concept of host/prey refuge, where an organism obtains protection from predation by hiding. Here, the term refuge is used as a generalisation, referring to spatial features that allow increased survival of individuals, which, in our model, correspond only to parasitoids finding alternative hosts and resources at these locations. Henceforth, we use refuge as reference to parasitoid refuge in this context.

At the beginning of each simulation, a fixed fraction of the patches is assigned as refuge areas, while the rest remains as crop areas. We have chosen three spatial arrangements to model common patterns of non-crop vegetation in agricultural landscapes (Fig 1): (i) random, where a given fraction of refuge areas are randomly and homogeneously distributed on the lattice, (ii) block, where the refuge areas are grouped in a square cluster located at the centre of the lattice, and (iii) border, where refuge areas are randomly chosen at the perimeter of the

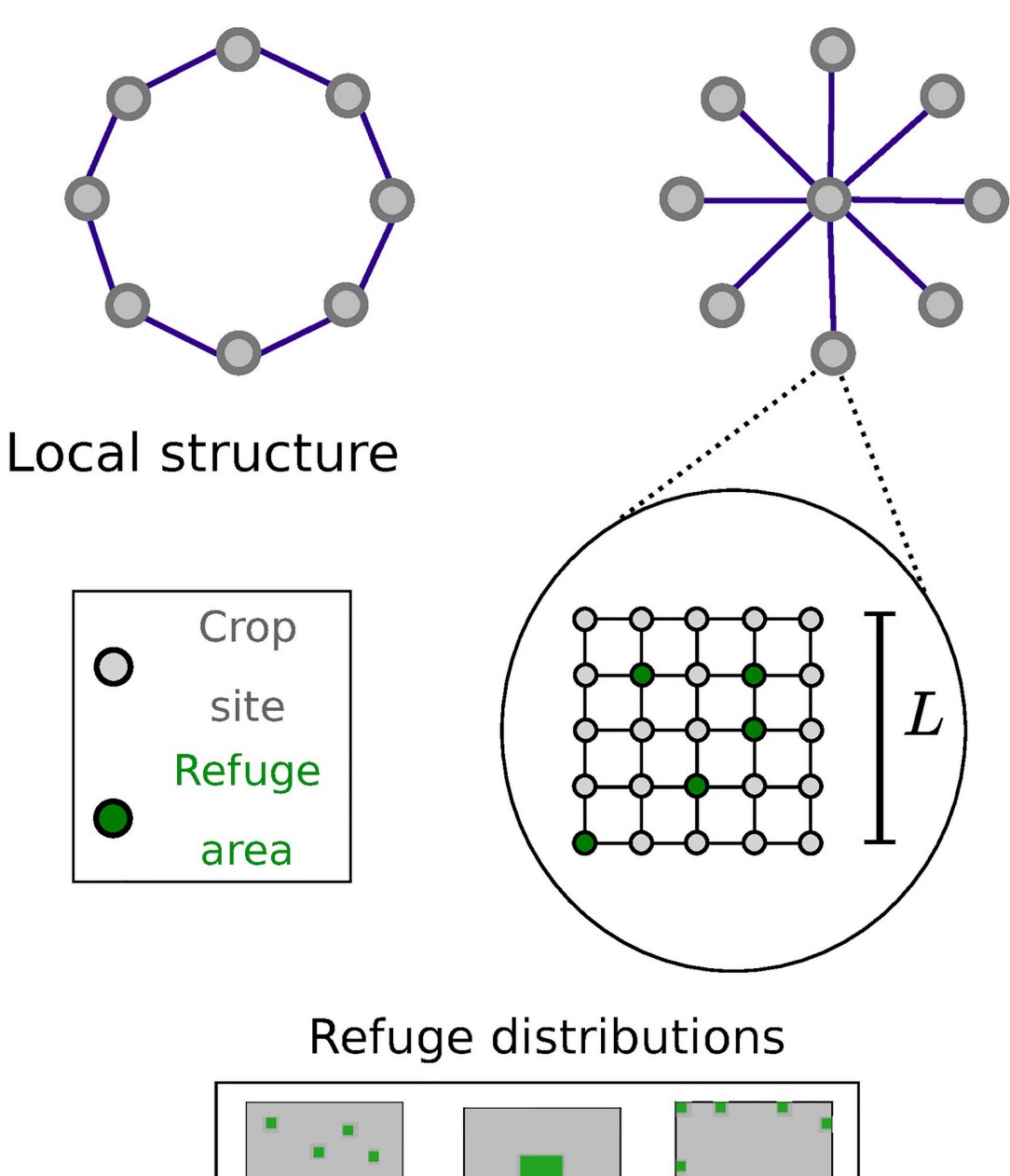

**Fig 1. Local and regional spatial structures.** Illustration of spatial structuring in the model, showing two different scales. On the regional scale, connectivity is represented by an external network, where network nodes are connected as ring or star topologies. Each of these nodes corresponds to a regular square lattice on the local scale, with patches assigned two types: crop or refuge (non-crop) areas. For each proportion of refuge areas on the lattice, three refuge distributions are considered: random, block, and border.

lattice. The different types of patches, crop or refuge, present different local dynamics for each species, as described below.

## Dynamics: Interaction

For the crop patches, interaction is given by a modified form of the host-parasitoid model proposed by Nicholson and Bailey [33], with parasitoids that attack hosts' eggs and host density-dependent effects that become relevant at the larval stage. This formulation is relevant, for example, for the interaction between lepidopteran pests and their parasitoids, but not limited to them.

Male parasitoids are not explicitly included in the model, therefore the female qualifier will be omitted when referring to parasitoid populations. If $H_{i,t-1}$ and $F_{i,t-1}$ are the densities of hosts and parasitoids at patch $i$ of the lattice at the previous generation (after dispersal step), the post-interaction host and parasitoid densities, $h_{i,t}$ and $f_{i,t}$, will be given by:

$$
\begin{aligned}
h_{i,t} &= \frac{\lambda k}{k + (\lambda - 1)H_{i,t-1}e^{-\alpha F_{i,t-1}}} H_{i,t-1}e^{-\alpha F_{i,t-1}} \\
f_{i,t} &= cH_{i,t-1}(1 - e^{-\alpha F_{i,t-1}}),
\end{aligned}
\tag{1}
$$

where $\lambda$ represents host growth rate, $k$ its carrying capacity, and $c$ the average number of parasitoid offspring emerging from each host. The term $e^{-\alpha F_{i,t-1}}$ represents the fraction of hosts that escapes parasitoid attack and $\alpha$ gives the parasitoid attack rate.

Host dynamics can be written as

$$
h_{i,t} = \lambda g(H_{i,t-1}, F_{i,t-1})H_{i,t-1}e^{-\alpha F_{i,t-1}}
$$

where the function $g(H_{i,t-1}, F_{i,t-1}) = \dfrac{k}{k + (\lambda - 1)H_{i,t-1}e^{-\alpha F_{i,t-1}}}$ corresponds to the limitation of growth by density-dependent effects, analogous to the one initially proposed by Beverton and Holt [34]. The inclusion of the term $e^{-\alpha F_{i,t-1}}$ on the function $g(H_{i,t-1}, F_{i,t-1})$, the main difference with the Berveton-Holt model, assumes an idea proposed by May [35]: that the number of hosts in the function for density dependence should be the one surviving parasitoid attack to the egg stage, therefore $H_{i,t-1}e^{-\alpha F_{i,t-1}}$. The proposed formulation takes into account specific details of the interaction, without explicitly considering the life stages of the insects.

For the patches representing refuge areas, the interaction dynamics is given by:

$$
\begin{aligned}
h_{i,t} &= 0 \\
f_{i,t} &= \frac{\lambda_{RA}k_{RA}}{k_{RA} + (\lambda_{RA} - 1)F_{i,t-1}} F_{i,t-1}
\end{aligned}
\tag{2}
$$

The subscript $RA$ stands for refuge area. We assume hosts as specialist pests for the resources on crop areas. Therefore, host individuals that arrive at refuge site $i$, after a dispersal event on the previous generation, die due to the lack of resources. Parasitoids on the other hand are generalists and therefore able to survive and reproduce (with small growth rate $\lambda_{RA}$ and carrying capacity $k_{RA}$) by using alternative hosts.

If the densities fall below 1.0, they are set to 0.0 characterizing an event of local extinction. That site can be recolonized at a posterior step time.

## Dynamics: Dispersal

Dispersal happens after the update to local populations due to the interaction. For each patch $i$ a given fraction of hosts will disperse to neighboring patches. This fraction depends on the local density of conspecifics. Following Reigada et al. [36], the density of dispersing hosts leaving patch $i$ at generation $t$ is:

$$h_{i,t}^{out} = \frac{\mu_H h_{i,t}^2}{h_{i,t} + h^0} \tag{3}$$

where $\mu_H$ is the maximum host dispersal rate in patches with large densities and $h^0$ relates to the tolerance to conspecifics. For small values of $h^0$, $h_{i,t}^{out}$ tends to $\mu_H h_{i,t}$, with dispersing hosts tending to the maximum fraction. For large values of $h^0$, $h_{i,t}^{out}$ tends to 0 with no dispersing hosts per site.

Neighboring patches on the lattice can be reached by hosts with a dispersal radius $R_H$ centred on patch $i$. Besides, hosts can also disperse to adjacent lattices if the two lattices are connected on the external network. If dispersal happens to another lattice, a destination patch is randomly chosen on the border of the neighbor lattice for the hosts to arrive.

Dispersal is stochastic, with a subtraction of a unit of density to the density of dispersing hosts at each step. Each unit is sent to one of the patches inside the dispersal radius with probability $P_j = \frac{C_i^H}{r_{ji}}$, where $C_i^H$ is a normalization constant. Dispersal to neighboring lattices is given with probability $P_j = \frac{C_i^H}{R_L}$, with $R_L$ representing the characteristic distance between nodes on the external network. The number of dispersal steps for each patch $i$ is given by the integer part of $h_{i,t}^{out}$ (Eq 3). If $h_{i,t}^{out} < 1$ no dispersal is implemented. After all patches $i$ are accounted for on all lattices, host populations are updated.

Parasitoid dispersal is implemented in an analogous fashion, with a slight difference depending on the type of the local patch. For crop patches $i$, the density of dispersing parasitoids at generation $t$ is given (according to Reigada et al. [36]) by:

$$f_{i,t}^{out} = \mu_F \frac{H^0}{H_{i,t} + H^0} \frac{f_{i,t}^2}{f_{i,t} + f^0} \tag{4}$$

but only if $H_{i,t} > 0$ (host density on site $i$ after dispersal stage). If $H_{i,t} = 0$ all the local parasitoid population disperses, thus $f_{i,t}^{out} = f_{i,t}$. Parameter $\mu_F$ gives the maximum parasitoid dispersal for small host densities and large parasitoid densities, while parameters $H^0$ and $f^0$ represent the necessary number of hosts to keep parasitoid at the patch, and the tolerance to parasitoid conspecifics, respectively.

For the refuge patch $i$, the density of dispersing parasitoids at generation $t$ is:

$$f_{i,t}^{out} = \mu_F \frac{f_{i,t}^2}{f_{i,t} + f^0} \tag{5}$$

Parasitoids disperse within a radius $R_P$, according to a similar process as the one for hosts, with neighboring patches being chosen with probability $P_j = \frac{C_i^P}{(r_{ji})^2}$, where $C_i^P$ is a normalization constant. We assume stronger distance limitation on parasitoid dispersal, thus the squared distance on the denominator. Neighboring lattices can be reached with probability $P_j = \frac{C_i^P}{(R_L)^2}$.

Again, no dispersal step is implemented if $f_{i,t}^{out} < 1$. All parasitoid populations are updated after all sites $i$ on all lattices are visited. Spatial population dynamics simulations were implemented in FORTRAN.

## Parameterisation

We investigated two regional connectivity arrangements, ring and star networks, and three different distributions of refuge areas, random, block and border. A number $N = 10$ of nodes on the external network (regional connectivity) was considered for all the scenarios. For the square lattice, we fixed $L = 33$, resulting in a total $L$ x $L = 1089$ patches for each lattice (if each patch has 10–100 m$^2$, each square lattice has approximately 1–10 ha). We also assumed reflective boundary conditions for the square lattices. Six values of occupation fractions for refuge areas were considered in the lattices: 0.008, 0.023, 0.045, 0.074, 0.111, 0.155, and 0.207 (corresponding to 9, 25, 49, 81, 121, 169, and 225 patches), distributed according to one of the three arrangements. For the border distribution refuge area fractions were considered up to 0.111 (corresponding to an almost completely covered perimeter of the square lattice). Although all the analyses were done with the fractions of refuge areas, in the Results section (section) we chose to show them as percentages to improve clarity.

Parameters for the interaction and dispersal stages, in particular maximum dispersal rates and tolerances of hosts and parasitoids to conspecifics at the same patch (see complete description on Methods), were obtained from a previous host-parasitoid experimental setting [37]. Since we were interested in the long-term behaviour of the system, attack rate and initial densities were adjusted to guarantee species coexistence in the majority of the simulations. All simulations start with initial densities of 300 hosts and 4 parasitoids, placed in one of the crop patches. For the star network, individuals are initially placed at the central node of the star. For the ring network, the node index for the initial placement is irrelevant, since all nodes are identical in connectivity structure. Parasitoids have their dispersal radius parameterised according to short ($R_P = 1$) or long ($R_P = 3$) range dispersal. The set of parameters is summarized in Table 1.

## Variables of interest

The effects of landscape structure and parasitoid dispersal ranges on host dynamics were evaluated on two variables: the average host density per crop site (total host density on each lattice divided by the number of crop sites) and the average host occupancy per crop site (number of sites occupied by hosts divided by the number of crop sites on the lattice). Simulations were iterated for $T = 5000$ generations. Average host density and occupancy were calculated over the last 100 generations for each simulation. The value $T = 5000$ corresponds to a very conservative choice, since fewer than 100 initial steps are typically needed to reach average values of host densities at each node of the external network. This choice, however, ensures no initial transients affect the patterns we describe (see S1 Fig in S1 File for examples of typical time series of host and parasitoid densities). For each fraction of refuge areas considered, 50 sample simulations were run with the same set of parameters.

We also assessed how spatial structure and parasitoid traits influence time variation on host dynamics. We calculated the Coefficient of Variation ($CV = s/\bar{x}$; ratio between the standard deviation, $s$, and the mean, $\bar{x}$, of a given sample) for the last 100 generations of the host density time series. We obtained one estimate of the CV for each of the 50 sample simulations, for each fraction of refuge areas and each value of parasitoid dispersal radius. The CV expresses how much the densities in a given interval deviate from the mean density for that interval, and can be used to compare scenarios with different means. Thus, we use it as a proxy to measure

**Table 1. Parameter values for the spatially-explicit simulation model.**

| Parameter name | Parameter description | Values** |
|---|---|---|
| $\lambda$ | Host growth rate (crop site) | 1.5 |
| $k$ | Host carrying capacity (crop site) | 3000 |
| $c$ | Average number of parasitoid offspring emerging from each host | 1 |
| $\alpha$ | Parasitoid attack rate | 0.05 |
| $\lambda_{RA}$ | Parasitoid growth rate (refuge area) | 1.01 |
| $k_{RA}$ | Parasitoid carrying capacity (refuge area) | 100 |
| $\mu_H$ | Maximum host dispersal rate (large host density) | 0.85 |
| $h_0$ | Host tolerance to conspecifics | 100 |
| $R_H$ | Host dispersal radius (within regular lattice) | 3* |
| $\mu_F$ | Maximum parasitoid dispersal rate (large parasitoid and small host densities) | 0.4 |
| $H_0$ | Host density threshold to prevent parasitoid dispersal from patch | 300 |
| $f_0$ | Parasitoid tolerance to conspecifics | 100 |
| $R_P$ | Parasitoid dispersal radius (within regular lattice) | 1*, 3* |
| $R_L$ | Distance between nodes on external network (regional distance) | 6* |
| $L$ | Square lattice size | 33 ($L^2 = 1089$) |
| $N$ | Number of nodes on external network | 10 |

* Lattice unit distance.

** Densities and carrying-capacities in units of number of individuals.

strong deviations from the mean density, or the propensity of a given scenario for the occurrence of population outbreaks.

## Statistical analyses

In order to evaluate how the two dependent variables of interest, namely average host density per crop site and average host occupancy per crop site, would respond to changes in percentage and distribution of refuge areas, as well as to the network structure representing regional connectivity, we have performed linear regression analyses to evaluate significant differences in the slopes. The fraction of refuge areas was used as continuous independent variable with the three distributions of refuge areas (random, block, and border) as levels of a categorical variable. For the star network, an additional categorical variable representing the connectivity degree of nodes (peripheral and central) was also considered. Comparisons between slopes obtained from linear regressions were also performed for the coefficient of variation of host density.

For the comparisons, we have performed T-tests for statistical significance of the difference in the slopes obtained from the regressions, considering each parasitoid radius ($R_P = 1$ and $R_P = 3$) separately. Thus, for the ring network we have pairwise comparisons for the slopes for each categorical distribution of refuge areas, while for the star network, pairwise comparisons for the distributions are done for each category of connectivity separately. Homoscedasticity and normality of the residuals for the regressions were checked for consistency. All the statistical analyses were performed in R and the results can be found in the Supporting Information (see S1 and S2 Tables in S1 File).

## Results

### Ring network

Fig 2 shows the results for the average host density per crop site (Fig 2—first row) and the average host occupancy per crop site (Fig 2—second row) for the ring network, as functions of the

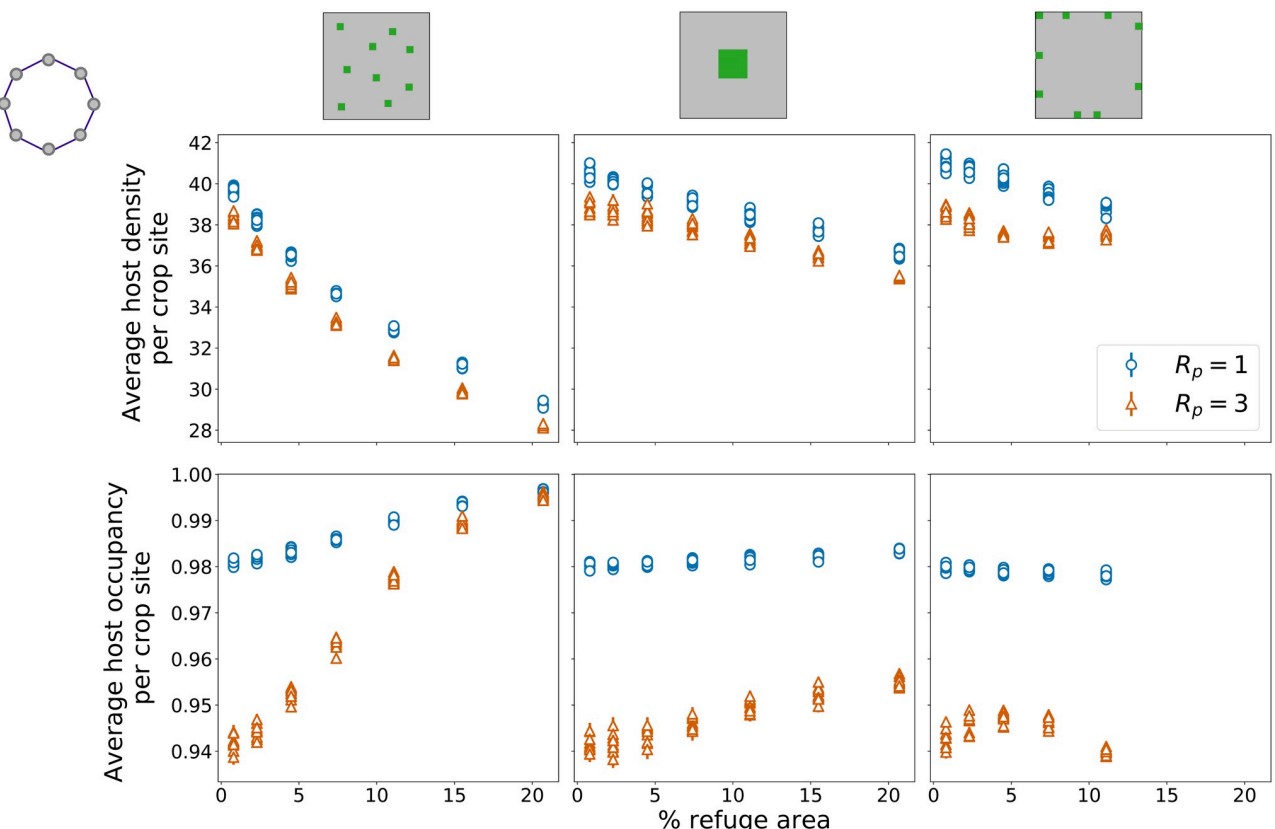

**Fig 2. Host variables—Ring network.** Average host density per crop site (first row) and average host occupancy per crop site (second row) as functions of the percentage of refuge areas on the lattice, for random, block, and border distributions of refuge areas (first, second, and third columns, respectively). Ten points are shown for each percentage of refuge areas, one for each node on the external network. Each point corresponds to the average density or occupancy for that node over 50 simulations, with error bars showing one standard error of the mean. Results are shown for two values of parasitoid dispersal radius $R_P = 1$ (blue circles) and $R_P = 3$ (orange triangles). External network (regional connectivity) has ring topology.

percentage of refuge areas on the square lattices. For each percentage of refuge areas, results were obtained for two parasitoid dispersal radii $R_P = 1$ and $R_P = 3$. Ensemble averages are shown with error bars, corresponding to one standard error of the mean. For each set of parameters, 50 samples were simulated. Densities and occupancies for the 10 nodes of the external network are shown, with 10 circles ($R_P = 1$) and 10 triangles ($R_P = 3$) for each value of percentage of refuge areas.

For all the scenarios, the 10 points for each fraction of refuge areas present very similar values, showing no significant difference in host variables among different nodes of the external network. This result follows directly from the topological equivalence between nodes on the ring network, with enough time given to populations to fully spread through the network (averages were calculated for the last 100 generations of the time series). Values for density and occupancy of hosts for $R_P = 3$ are always lower than the values obtained for $R_P = 1$.

For all the distributions of refuge areas, the average density of hosts per crop site decreases as the fraction of refuges increases, with the largest reduction obtained for the random distribution (Fig 2, top left panel). The decrease in host density for the random distribution, however, is followed by the largest increase in average occupancy of hosts per crop site on the lattice (Fig 2, bottom left panel). This increase does not represent a simple spread of host

individuals on the lattice, since the total lattice densities (host density per site times number of crop sites) decrease when the fraction of refuge areas goes from 0.008 to 0.207.

Each distribution of refuge areas determines different responses of the dependent variables as the proportion of refuge areas increases. We show significant differences on the slopes of the regression curves for both host density and host occupancy, comparing the three refuge distributions in pairs (S1 Table in S1 File), for both values of parasitoid dispersal ranges ($R_P$ = 1 and $R_P$ = 3). The only exception are the slopes for host density of block and border distributions for $R_P$ = 1 (Fig 2, top central and top right panels) which are not significantly different ($p$ = 0.4043—S1 Table in S1 File).

## Star network

Fig 3 shows the effect of the distributions of refuge areas on host density and occupancy as refuge area increases, for both parasitoid dispersal ranges ($R_P$ = 1 and $R_P$ = 3) for the star network. Host density and host occupancy show a clear separation of the values in two distinct categories: the peripheral nodes of the external network (empty symbols) and the central node (filled symbols). The central node presents consistently lower densities and higher occupancies compared to peripheral nodes, for each percentage of refuge areas on all the scenarios.

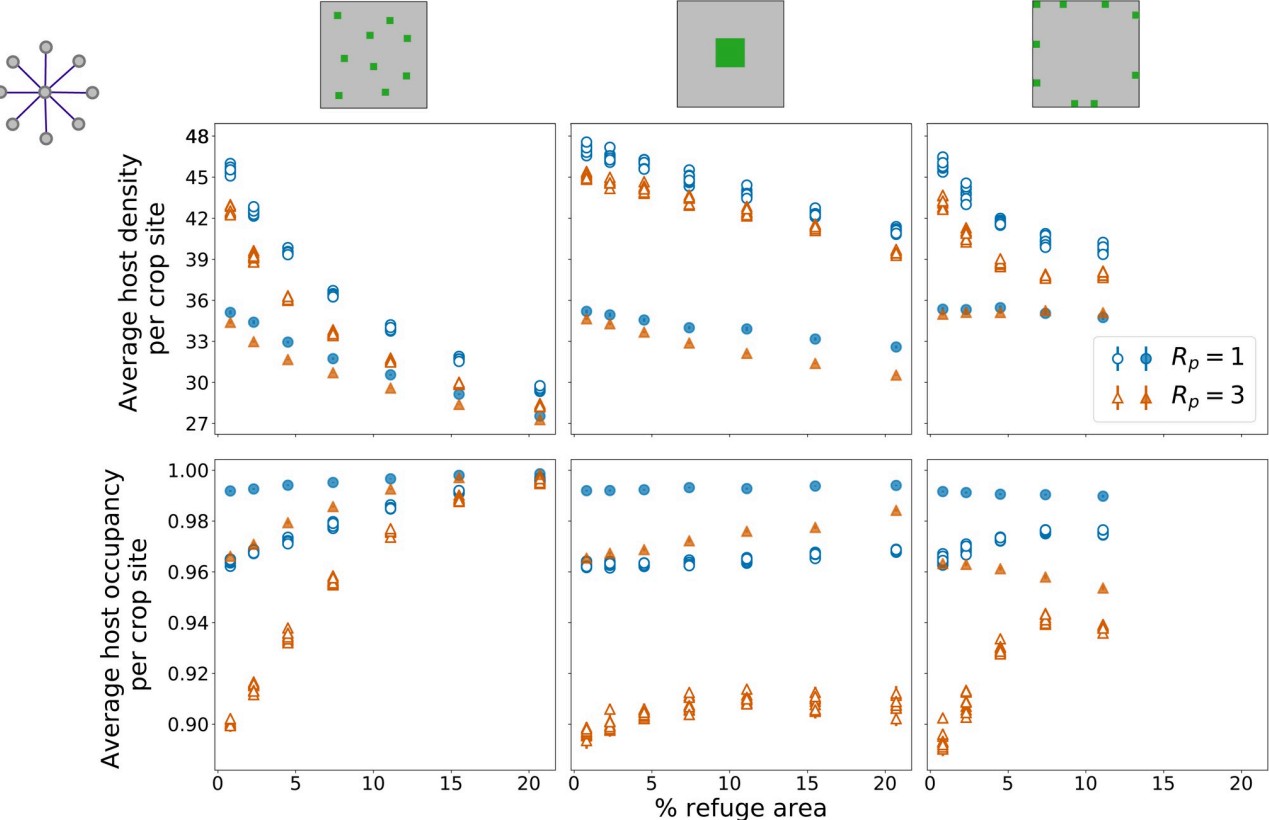

**Fig 3. Host variables—Star network.** Average host density per crop site (first row) and average host occupancy per crop site (second row) as functions of the percentage of refuge areas on the lattice, for random, block, and border distributions of refuge areas (first, second, and third columns, respectively). Ten points are shown for each percentage of refuge areas, one for each node on the external network. Peripheral nodes are represented as empty symbols and central nodes as filled symbols. Each point corresponds to the average density or occupancy for that node over 50 simulations, with error bars showing one standard error of the mean. Results are shown for two values of parasitoid dispersal radius $R_P$ = 1 (blue circles) and $R_P$ = 3 (orange triangles). External network (regional connectivity) has star topology.

A reduction of the difference between peripheral and central nodes as the fraction of refuge areas increases can be seen, in particular for the random and border distributions (both for density—Fig 3, top left and top right panels—and occupancy—Fig 3, bottom left and bottom right panels). For the block distribution, this reduction in the differences between node categories is less pronounced and shows a slight increase for host occupancy at $R_P = 3$ (Fig 3, bottom central panel).

The effects of refuge distributions on the responses of host densities and host occupancies to the percentage of refuge areas are statistically significant, separately compared for each parasitoid radius and node category (S2 Table in S1 File). The exceptions are the difference between block and border distributions on host density, for central nodes and $R_P = 1$, and the difference between random and border on host occupancy, for peripheral nodes and $R_P = 3$, which present large confidence intervals on the estimation of slopes.

The separation between central and peripheral nodes reveals an important aspect that could not be observed on the ring network. For the ring network, the border distribution presents a small variation in host density and occupancy, as the fraction of refuge areas increases (Fig 2, top right and bottom right panels). For the same distribution of refuge areas on the star network, there is a clear decrease in density and increase in occupancy for the peripheral nodes as the percentage of refuge areas increases, while the central nodes present no clear variation (Fig 3, top right and bottom right panels). Thus, the connectivity of this lattice on the regional scale influences how host densities and occupancies will be affected by local percentages of refuge areas. For other refuge distributions (random and block) in star networks, host densities decrease and host occupancies increase as refuge areas increase, for peripheral and central nodes, with significant differences in these tendencies when refuge distributions are compared (S2 Table in S1 File).

Values for host density and occupancy corresponding to the parasitoid dispersal radius $R_P = 3$ are always lower than the ones obtained for $R_P = 1$ (Fig 3), if comparisons are made between nodes on the same category of regional connectivity (peripheral nodes with $R_P = 1$ compared to peripheral nodes with $R_P = 3$, for example).

## Coefficients of variation (CV) of host density

Fig 4 shows the results for the ensemble averages of the CV for host density per crop site, for random, block and border distributions of refuge areas, in ring and star external networks. Our results show that the interplay between regional connectivity and spatial distribution of refuges influences the variability in relation to the mean density (measured by the CV) for the majority of the scenarios, with significant differences in regression slopes when comparing pairs of distributions of refuge areas (S1 and S2 Tables in S1 File). Important exceptions are the comparisons between block and border, for the ring network with $R_P = 3$, and between block and border in the star network, for both central and peripheral nodes, with $R_P = 1$, for which the differences in slopes are not significant.

For the random distribution, an increase in the percentage of refuges leads to a decrease in the CV for both ring and star networks. The variation on the CV is not as evident for block and border distributions. For the star network, however, a clear separation in the values of CV for peripheral and central nodes (empty and filled symbols, respectively) is also shown, with central nodes presenting consistently lower values of the CV, for every percentage of refuge areas. For the border distribution in the star network (Fig 4, bottom right panel), an initial decrease in CV can also be noted at peripheral nodes, stabilising after 5% of refuge areas. For the central nodes, the CV increases following an increase in refuge areas, leading to a reduction on the separation between peripheral and central nodes.

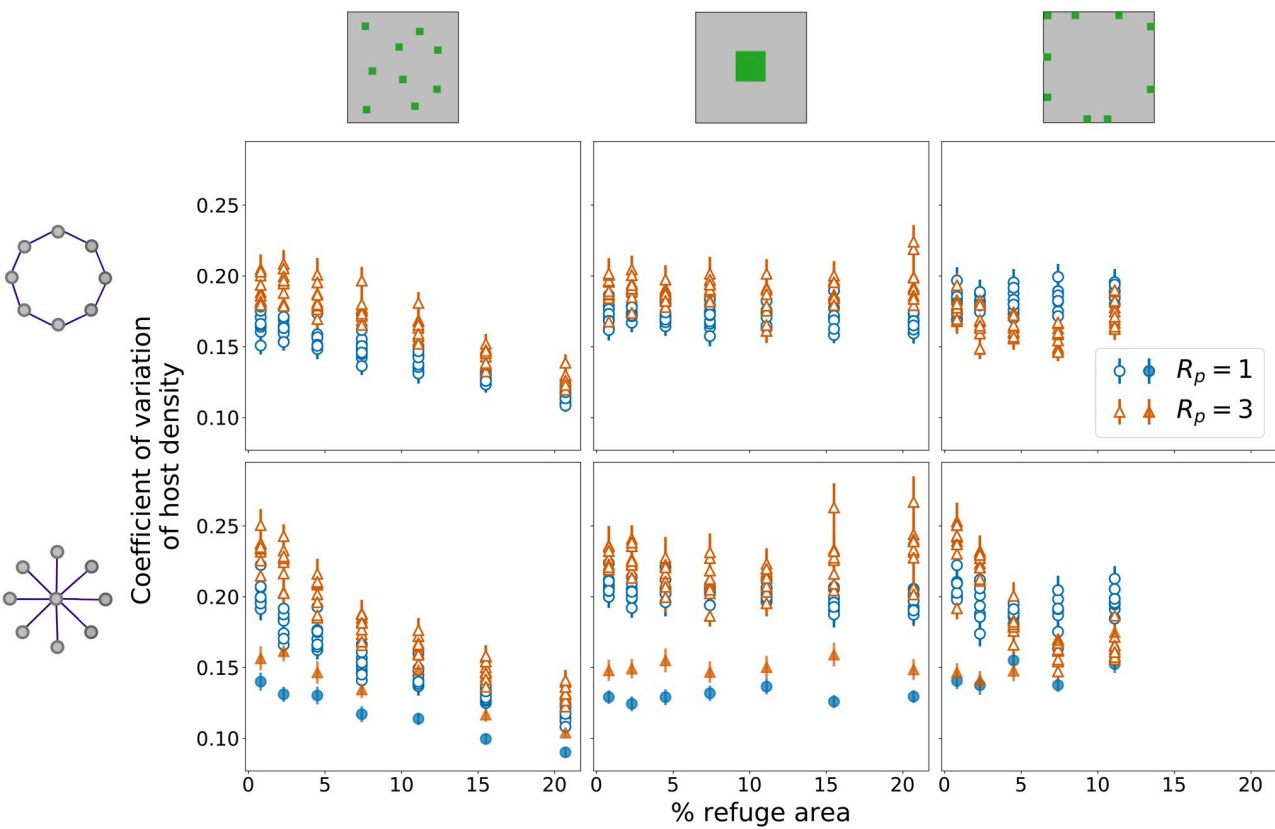

**Fig 4. Coefficients of variation—Ring and star networks.** Coefficients of variation (CV) for host density per crop site as a function of the percentage of refuge areas for the ring (first row) and star (second row) networks with random, block, and border distributions of refuge areas (first, second, and third columns, respectively). Ten points are shown for each percentage of refuge areas, one for each node on the external network. For the star network, peripheral nodes are represented as empty symbols and central nodes as filled symbols. Each point corresponds to the average CV for that node over 50 simulations, with error bars representing one standard error of the mean. Results are shown for two values of parasitoid dispersal radius $R_P = 1$ (blue circles) e $R_P = 3$ (orange triangles).

The values of the CV for $R_P = 3$ are slightly greater than the ones for $R_P = 1$ in all of the scenarios, except for the border distribution in the ring network (Fig 4, top right panel), and for larger fractions of refuge areas for the border distribution in the star network (Fig 4, bottom right panel).

## Discussion

Understanding how the structural characteristics of the landscapes affect species dynamics and distribution is fundamental to devise better strategies for pest control and conservation. Landscape composition has been widely recognised to have an important effect on biological control in agroecosystems, specifically in terms of proportion and distribution of non-crop or semi-natural habitats, by providing shelter and alternative resources for natural or introduced enemies [14, 22, 38, 39]. Moreover, it has also been shown that different scales of landscape structure from local to regional level are important to understand pest dynamics and distributions in natural and managed environments [7, 8, 40]. Our main goal was to establish how these two scales of spatial organisation interact and determine different responses to population dynamics of interacting species.

We used a host-parasitoid mathematical model to investigate the influence of spatial structure on population dynamics, focusing primarily on host density and occupancy on the

landscape. We explored two different spatial scales: at the local scale, analysing different proportions and distribution of refuge areas, and at the regional scale, different patterns of connectivity on the landscape. Parasitoid dispersal radius, another relevant parameter for the host-parasitoid interaction, was also varied, showing how the interplay between biotic and abiotic aspects might produce both qualitative and quantitative changes on the observed patterns. Our results reveal three non-intuitive key findings about the role played by refuge areas on species dynamics.

First, we show that hosts' temporal and spatial dynamics are dependent not only on the proportion of parasitoid refuges on the landscape, but also on their distribution and on the connectivity structure at the regional scale. In our model, refuge areas correspond to non-crop patches where it is possible to find alternative hosts and food supplies, allowing reproduction and maintenance of a small population of parasitoids [22]. Therefore, it was expected that an increase in the number of refuge sites (viewed as landscape sources of parasitoids) would always be associated with decreases in density and occupancy of hosts at crop sites [41]. However, our results show that decreases in density and occupancy are not always the rule, with significant differences on the response of these variables to refuge proportions, depending on the spatial distributions of refuges.

Second, the results we have obtained show how host populational patterns are dependent on the parasitoid dispersal radius, but also how spatial structure plays a fundamental role. While in most cases an increased parasitoid dispersal radius leads to decreased values for host density and occupancy and increased values for coefficients of variation, changes to this general expectation can be seen for specific spatial arrangements (see border distribution in Fig 4, top right and bottom right panels). This fact suggests that the choice of agents for biological control should take into account not only dispersal or interaction traits, but also the landscape structure where this control is meant to be applied [42].

Third, the difference on the patterns observed for ring and star networks shows that the same distribution of refuge areas at the local scale may present meaningful differences on the dynamical effects depending on the landscape connectivity at the regional scale. The separation on the host population variables related to peripheral and central nodes for the star network highlights an important interplay between local distribution of refuge areas and landscape connectivity, since the more connected nodes in the regional scale (central nodes on the external networks) consistently present smaller values for host density and larger values for host occupancy. For the ring network, there was no significant distinction between patterns observed for each node.

In an agricultural context, where hosts could represent insect-pests for crop areas, our results might inform design strategies for biological control, through the analysis of landscape connectivity and restructuring of refuges to sustain natural or introduced populations. With units of production located on a regional landscape with different degrees of connectivity, biological control of pests would benefit of an integrated regional approach [43, 44]. In this scenario, our results suggest that the identification of central and peripheral nodes, might help to articulate directed actions on key units, such as engineering refuge distributions or augmented biological control.

Comparing the three refuge distributions, the random distribution presented the largest reduction on host density as a function of the proportion of refuge areas, in agreement to the beneficial effect of high edge density on pest control in real landscapes [10]. The choice of this refuge distribution as the best candidate for pest control in the context of our modelling approach, however, must be treated with caution. As a simplifying assumption, we have not considered possible variations of population growth of parasitoids at refuge sites. At these sites, we assume logistic growth for parasitoids, with population stochasticity only due to

dispersal. A complex population model could be obtained, for example, by considering the dynamics of alternative hosts at refuges. This would have a relevant effect on the dynamical response of populations subject to the random distribution of non-crop areas, particularly for small fractions of refuges on the lattice, since these sites would be fairly isolated. In this scenario, the assumption that all of them could constitute, at any time, refuges capable of supporting stable communities of alternative hosts would no longer hold. Thus, the dynamics of alternative hosts at refuge areas constitutes an important aspect to be addressed in future work.

We show a clear distinction in effectiveness of host density reduction, measured by differences in slopes for different spatial distributions of parasitoid refuges. This relates to a long running discussion regarding the relevance of refuge and native area fragments on the maintenance of natural enemies populations and biodiversity conservation in general [45–48], specially regarding the effects of size and distribution of these fragments. Although our work hints to the relative role of fragmentation for biological control on agroecosystems, a complete evaluation of the matter would also consider the dispersal patterns of the species under study [49]. Nevertheless, our results clearly present a separation between the percentage of refuge habitats and their spatial distribution, which has also been the focus of much recent debate [50, 51].

Finally, although we have focused essentially on the population variables of hosts, understanding the reciprocal actions of these variables with the population dynamics of parasitoids is extremely important to comprehend the biotic and abiotic factors that condition the spatio-temporal species patterns in this context. In particular, future work should address how refuge distributions and regional connectivity influence the occurrence of population outbreaks for both species in time and space. The study of these interrelations can provide valuable knowledge to inform policy on pest control and spatial structuring of these landscapes.

## Supporting information

**S1 File. Supporting information with time series for host and parasitoid densities, for different distributions of refuge areas (S1 Fig) and summary statistics for ring and star networks (S1 & S2 Tables).**
(PDF)

## Author Contributions

**Conceptualization:** Lucas D. Fernandes, Wesley A. C. Godoy, Carolina Reigada.

**Data curation:** Lucas D. Fernandes, Angelica S. Mata.

**Formal analysis:** Lucas D. Fernandes, Angelica S. Mata, Carolina Reigada.

**Funding acquisition:** Wesley A. C. Godoy.

**Investigation:** Lucas D. Fernandes, Angelica S. Mata.

**Methodology:** Lucas D. Fernandes, Angelica S. Mata, Wesley A. C. Godoy, Carolina Reigada.

**Project administration:** Lucas D. Fernandes, Wesley A. C. Godoy.

**Resources:** Angelica S. Mata, Wesley A. C. Godoy.

**Software:** Lucas D. Fernandes, Angelica S. Mata, Carolina Reigada.

**Supervision:** Wesley A. C. Godoy, Carolina Reigada.

**Validation:** Lucas D. Fernandes.

**Visualization:** Lucas D. Fernandes, Carolina Reigada.

**Writing – original draft:** Lucas D. Fernandes, Angelica S. Mata, Wesley A. C. Godoy, Carolina Reigada.

**Writing – review & editing:** Lucas D. Fernandes, Angelica S. Mata, Wesley A. C. Godoy, Carolina Reigada.

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
