## [Decision Letter · Decision Letter 0]

9 Dec 2021

PONE-D-21-33487Refuge distributions and landscape connectivity affect host-parasitoid dynamics: motivations for biological control in agroecosystemsPLOS ONE

Dear Dr. D. Fernandes,

Thank you for submitting your manuscript to PLOS ONE. After careful consideration, we feel that it has merit but does not fully meet PLOS ONE’s publication criteria as it currently stands. Therefore, we invite you to submit a revised version of the manuscript that addresses the points raised during the review process.

Please address carefully all the points raised by the reviewers. In particular, please highlight the novelty of your model compared to others and make a more explicit connection to realistic ecological scenarios.

We look forward to receiving your revised manuscript.

Kind regards,

Alejandro Carlos Costamagna, Ph.D.

Academic Editor

PLOS ONE

Journal Requirements:

Additional Editor Comments (if provided):

I only have a couple of minor comments. Figure 2-3: are SE present but very small? Please confirm. Line 338: replace “Parasitoids” by “Parasitoid”.

Reviewers' comments:

Reviewer's Responses to Questions

**Comments to the Author**

1. Is the manuscript technically sound, and do the data support the conclusions?

Reviewer #1: Yes

Reviewer #2: Yes

2. Has the statistical analysis been performed appropriately and rigorously? 

Reviewer #1: I Don't Know

Reviewer #2: Yes

3. Have the authors made all data underlying the findings in their manuscript fully available?

Reviewer #1: Yes

Reviewer #2: Yes

4. Is the manuscript presented in an intelligible fashion and written in standard English?

Reviewer #1: Yes

Reviewer #2: Yes

5. Review Comments to the Author

Reviewer #1: I definitely see the relevance of thinking of different spatial scales of landscape, and scales of movement influencing host-parasitoid interactions, their abundances and dynamics. But I don’t see how the scales presented here correspond to agricultural landscapes. Figure 1 shows small scale areas with refuge imbedded in it (individual farms), connected together on a larger in ring and star configurations. What does the area between the farms represent? Why that configuration? What is the difference between that and “non-crop” habitat which is called refuge when it is in the farm?

Since there are more important generalist predators than parasitoids in agricultural systems, it may make sense to change the language in the ms to be about predator-prey rather then host-parasitoid dynamics.

I am not qualified to evaluate the novelty of the model, but I will note that it wasn’t discussed in relation to other spatial host-parasitoid dynamics models. Thus maybe it is difficult to assess the contribution of this particular model and set of simulations. Perhaps analysis of local and regional scales simultaneously is novel, but there has been significant modeling of the two scales separately. The value of this study would be increased by placing it more the context of other work.

Overall, I found the introduction and abstract unclear, but the results and discussion pretty clear. Though the discussion was clear in could benefit from more reference to related work from the literature.

Specific comments:

Abstract:

-Consider leaving out the second sentence

-What do you mean by non-crop habitat? Do the hosts not inhabit it? Later I see what you mean, but in the abstract it is confusing.

-The abstract will be more interesting with specific outcomes of the study stated, so what are some “responses of host suppression”? And, how does parasitoid dispersal range interact with landscape structure to affect host population variables?

Line 15-16: If you are sticking with the host-parasitoid set up rather than predator-prey, consider introducing the work in the context of other spatial models of host-parasitoid dynamics such as C. Briggs, W. Murdoch, J. vandermeer, P. Amarasekare, Susan Harrison

Line 20, references 16 and 17: Again, these don't seem like specific references for this idea.

Line 23 references: These references are more about predators than parasitoids.

Line 30: missing the word "viable"?

Line 38: “Guarantee” sounds pretty absolute. How about facilitate? or promote?

Line 41: Connectivity is defined by different people in different ways. Sometimes connectivity is a function of the distribution of the habitat and the dispersal ability of the species, so it isn't independent of dispersal.

Line 43: Is two distinct scales a general pattern? I don't quite see it. I can imagine movement within a patch that isn't really dispersal, and then dispersal, that could be at a range of scales, but not two distinct ones. Am I missing something?

Line 47: Define what you mean by non-crop. How is that related to the term “refuge” as you use it?

Line 51: What are the host variables?

Line 53: What is connectivity in this case, and the dynamics of which species? And the spatial dynamics or population growth rate dynamics?

Line 58-59: I didn’t get from the introduction how these two discrete scales correspond to in real landscapes.

Line 61-63: This should be said earlier. Still though, I don't see how this is two scales. Is there refuge or crop or something else between farms?

Line 95: Density dependence of what?

Line 162-163: Why a star and a ring?

Line 175-176: Explain the Parameters briefly so that the reader understands generally without looking at another paper.

Line 186: What are these traits and how were they chosen?

Reviewer #2: This contribution explores the dynamics of host-parasitoid systems in connected patches based on different topologies. I think the paper is well written and the topic of spatial effects in quite pertinent to these systems. However, some discussion around these points is missing in the paper:

1) The concept of refuge has to be motivated and discussed. Generally, refuges imply a host refuges where parasitoids are unable to attack hosts, but I found this concept of “parasitoid refuge” confusing. What is the motivation for a parasitoid refuge and how is that different from host refuge?

2) Connected to the earlier point, host refuges are critical to stabilize host-parasitoid systems So the authors should include them in the simulations to see how that the overall dynamics and results are altered.

3) The authors mention dynamics but there no time-traces of population densities over time. It will be good include these to see what the actual dynamics looks like

4) The Nicholson Bailey models is unstable resulting in diverging oscillations. Generally, I found a lot of this context on stability missing in the entire papers. I could not find the stability analysis for these systems to see if they are indeed stable or not. Can the authors present this analysis?

5) This recent paper came up in my search on spatial effect and seems relevant to this study. It will be good discuss and compare

https://www.sciencedirect.com/science/article/abs/pii/S0025556420300869

6. PLOS authors have the option to publish the peer review history of their article (what does this mean?). If published, this will include your full peer review and any attached files.

Reviewer #1: No

Reviewer #2: No

---

## [Author Response · Author response to Decision Letter 0]

16 Feb 2022

Please, see response to editor and reviewers on the attached files ("response_to_reviewers" and "PlosOne_cover_letter_resubmission")

---

## [Editor Report · Decision Letter 1]

23 Mar 2022

PONE-D-21-33487R1Refuge distributions and landscape connectivity affect host-parasitoid dynamics: motivations for biological control in agroecosystemsPLOS ONE

Dear Dr. D. Fernandes,

Thank you for submitting your manuscript to PLOS ONE. After careful consideration, we feel that it has merit but does not fully meet PLOS ONE’s publication criteria as it currently stands. Therefore, we invite you to submit a revised version of the manuscript that addresses the points raised during the review process.

The authors have answered satisfactorily all the questions raised by the reviewers and the editor in this new version. The only minor change I request is to modify a sentence that currently does not make sense grammatically:

L 101-102: However, treating refuges as more general spatial features that allow increased survival of individuals (in this case, parasitoids), we use this term henceforth.

Please note also that the connection with the use of ‘refuge’ is lost, I think, by the previous sentence. Once you change this sentence, please resubmit the new version.

We look forward to receiving your revised manuscript.

Kind regards,

Alejandro Carlos Costamagna, Ph.D.

Academic Editor

PLOS ONE
---

## [Author Response · Author response to Decision Letter 1]

30 Mar 2022

We thank the reviewer for the thorough evaluation of our manuscript. We have corrected the previous sentence, better explaining our use of the term refuge, which now reads:

"Note that this is different from the more common concept of host/prey refuge, where an organism obtains protection from predation by hiding. Here, the term refuge is used as a generalisation, referring to spatial features that allow increased survival of individuals, which, in our model, correspond only to parasitoids finding alternative hosts and resources at these locations. Henceforth, we use refuge as reference to parasitoid refuge in this context."

---

## [Editor Report · Decision Letter 2]

1 Apr 2022

Refuge distributions and landscape connectivity affect host-parasitoid dynamics: motivations for biological control in agroecosystems

PONE-D-21-33487R2

Dear Dr. D. Fernandes,

We’re pleased to inform you that your manuscript has been judged scientifically suitable for publication and will be formally accepted for publication once it meets all outstanding technical requirements.

Kind regards,

Alejandro Carlos Costamagna, Ph.D.

Academic Editor

PLOS ONE
---

## [Editor Report · Acceptance letter]

6 Apr 2022

PONE-D-21-33487R2 

Refuge distributions and landscape connectivity affect host-parasitoid dynamics: motivations for biological control in agroecosystems 

Dear Dr. D. Fernandes:

I'm pleased to inform you that your manuscript has been deemed suitable for publication in PLOS ONE. Congratulations! Your manuscript is now with our production department. 

Kind regards, 

on behalf of

Dr. Alejandro Carlos Costamagna 

Academic Editor

PLOS ONE